# In Vivo and In Vitro Biological Evaluation and Molecular Docking Studies of Compounds Isolated from *Micromeria biflora* (Buch. Ham. ex D.Don) Benth

**DOI:** 10.3390/molecules27113377

**Published:** 2022-05-24

**Authors:** Abdullah S. M. Aljohani, Fahad A. Alhumaydhi, Abdur Rauf, Essam M. Hamad, Umer Rashid

**Affiliations:** 1Department of Veterinary Medicine, College of Agriculture and Veterinary Medicine, Qassim University, Buraydah 52571, Saudi Arabia; 2Department of Medical Laboratories, College of Applied Medical Sciences, Qassim University, Buraydah 52571, Saudi Arabia; f.alhumaydhi@qu.edu.sa; 3Department of Chemistry, University of Swabi, Anbar 23561, Pakistan; 4Department of Food Science and Human Nutrition, College of Agriculture and Veterinary Medicine, Qassim University, Buraydah 52571, Saudi Arabia; e.hamad@qu.edu.sa; 5Department of Chemistry, COMSATS University Islamabad, Abbottabad 22060, Pakistan; umerrashid@cuiatd.edu.pk

**Keywords:** *Micromeria biflora*, anti-inflammatory, analgesic, muscle relaxant, sedative, anti-inflammatory, anticancer, in vitro COX-1/2, in silico studies

## Abstract

*Micromeria biflora*, a traditional medicinal plant, is extensively used for treating various painful conditions, such as nose bleeds, wounds, and sinusitis. A phytochemical investigation of the chloroform fraction of *Micromeria biflora* led to the isolation of salicylalazine. Salicylalazine was assessed in vivo for analgesia, muscle relaxation, sedative, and anti-inflammatory properties, as well as in vitro for COX-1/2 inhibition activities. It was assessed against a hot plate-induced model at different doses. The muscle relaxant potential of salicylalazine was evaluated in traction and inclined screening models, while sedative properties were determined using an open-field model. The anti-inflammatory potential of salicylalazine was assessed in histamine and carrageenan-induced paw edema screening models. Salicylalazine exhibited significant analgesic potential in a dose-dependent manner. In both screening models, an excellent time-dependent muscle-relaxation effect was observed. Salicylalazine demonstrated excellent sedation at high doses. Its anti-inflammatory activity was determined through the initial and late phases of edema. It exhibited anticancer potential against NCI-H226, HepG2, A498, and MDR2780AD cell lines. In vitro, salicylalazine showed preferential COX-2 inhibition (over COX-1) with an SI value of 4.85. It was less effective in the initial phase, while, in the later phase, it demonstrated significant effects at 15 and 20 mg/kg doses compared with the negative control. Salicylalazine did not exhibit cytotoxicity in the MTT assay, preliminarily indicating its safety.

## 1. Introduction

Medicinal herbs are a rich source of chemical constituents; they produce diverse classes of phytochemicals precisely and accurately. The chemical constituents of medicinal herbs are responsible for the different biological efficacy of plants [1]. The majority (90%) of potential drug molecules can be extracted from medicinal plants directly or indirectly [2]. According to a WHO report, approximately 80% of African, as well as Asian countries, use herbal products for primary health care [3]. Up until now, only 1 percent of people among 0.26 million medicinal plants have been explored phytochemically and biologically. Natural products have biological potential and are present in various parts of the plants, including hardwood, leaves, roots, etc. [4]. *M. floraa* is a common member of the family Lamiaceae, which is commonly identified as white leaf savory and qurniyya. It is an evergreen shrub distributed in various regions of Saudi Arabia, Israel, the Himalayas, Pakistan, the Middle East, and the eastern Mediterranean region at altitudes ranging from 1000 to 3000 m. It is widespread throughout Asia, North America, Europe, and North America, with moderate diversity in the Mediterranean area as well as the Canary Islands. The plant is also found in the northern region of Pakistan, where it blooms from June to August, and the seeds ripen in August and September [5]. *M. biflora* pollinated over insects which are known as hermaphroditic [6]. Its leaves are aromatic and are used in the preparation of decoctions (herbal tea). In the folkloric system, *M. biflora* paste is prepared and used to cure toothache and as a poultice for wound healing. The rubbing of plants produces a fragrance that is inhaled to treat nose bleeding. Meanwhile, orally eating the plant’s juice is used for the efficient curing of sinusitis [7]. The essential oils isolated from *M. biflora* have been documented for their antipyretic, anti-inflammatory, and analgesic potentials [8]. Tea prepared from *M. biflora* is commonly used for treating throat irritation and flu, while its extracts are used to treat stomach disorders, pneumonia, and headaches [9]. Phytochemical studies on *M. biflora* led to the isolation of salicylalazine, which showed excellent urease, tyrosinase, phosphodiesterase, and antiglycation activities [9]. The current findings highlight the medicinal status of *M. biflora**,* which has been previously documented for the cure of different diseases. In the current study, the chloroform soluble extract was subjected to chromatographic analysis, yielding salicylalazine. Salicylalazine was assayed for in vivo analgesic, muscle relaxation, sedative, and anti-inflammatory properties, as well as in vitro COX-1/2-inhibitory and anticancer activities.

## 2. Results 

### 2.1. Characterization of Salicylalazine

Molecule 1 (m.p 215–218; purity 99.21) was purified from the chloroform fraction of *M. biflora* through repeated normal-phase column chromatography. Its HRMS (*m/z*) exhibited a molecular ion peak at 241.0970, showing that the molecular formula is C_14_H_12_N_2_O_2_.UV spectroscopy showed an absorption peak at 394 nm. IR spectra exhibited absorption peaks at 1625 for C=N, 983, N-N, 1160, C-C, 1315, and C-OH cm^−1^ starching. NMR spectroscopy showed a dimer. There are six sets of proton signals showing a downfield –OH δ_H_ 11.0 (1H, s) and -N=CH, 8.75 (IH, s) and equivalent protons H-6, H-16, 7.45 (each 1H, dd), H-1, and −15 7.38 (each IH, dd). Similarly, H-2 and H-14 are equivalent protons that showed signals of δ_H_ 7.07 (each 1H, dd), H3, and H-13, 6.98 (each (1H d). C13 NMR showed seven carbons with δ_C_ 164.7, 159.8, 133.5, 132.6, 119.7, 117.3, and 117.2. The structure of salicylalazine (Figure 1) was recently elucidated by our research group using nuclear magnetic resonance spectroscopy and mass spectrometry [9].

### 2.2. Analgesic Effect

Salicyalazine showed excellent (*p* < 0.001) central analgesic activities at different doses, including 2.5, 5, 10, 15, and 20 mg/kg (Table 1). Salicyalazine amplified the latency time from the start of administration of salicylamide and kept a significant (*p* < 0.001) increase up to 120 min. Salicyalazine did not exhibit any significant effects at the lower doses. 

The results are shown as the mean ± SEM of all animals’ tolerance to thermal stimuli in seconds.

### 2.3. Muscle Relaxation Effects

Salicyalazine was assessed for muscle relaxation potential on an inclined plane and traction model, which exhibited a significant muscle relaxation effect (Table 2). Salicyalazine showed dose- and time-dependent muscle relaxation properties. Salicyalazine showed excellent effects in both models. The effect of salicylamide was highly potent at the start of the experiment against the standard.

The obtained results are shown as the mean ± SEM of all animals’ tolerance to thermal stimuli in seconds.

### 2.4. Sedative Effects

The sedative effects of salicyalazine are presented in Table 3. The compound salicyalazine showed an excellent (*p* < 0.001) sedative potential in a dose-dependent manner. Salicyalazine at 20 mg/kg exhibited a promising effect (*p* < 0.001), as represented by delaying the movement of animals in a distinct box.

### 2.5. Anti-Inflammatory Effect

Our results showed that salicyalazine attenuated carrageen-induced paw edema (6.09%), which reached a maximum of 80.98% after 3 h at 15 mg/kg and remained excellent at the 5 h time-point. Similarly, a significant effect was observed at 20 mg/kg for salicyalazine, where it exhibited 45.11% inhibition after 1 h of the experiment, which reached a maximum of 90.22% 3 h after carrageenan administration. The compound showed mild activity at 2.5, 5, and 10 mg/kg compared to the higher dose. The standard (diclofenac sodium) showed a promising effect compared to salicylamide (Figure 2).

### 2.6. In Vitro Cytotoxicity Effect 

The in vitro cytotoxicity of salicylalazine isolated from *M. biflora* was identified using the MTT assay. Paclitaxel was used as a standard drug. The results of cytotoxicity effects are presented in Table 4. Salicylalazine exhibited excellent activity against selected cell lines.

### 2.7. In Vitro Inhibition of Cyclooxygenases

The in vitro inhibition of cyclooxygenases (COX-1 and COX-2) by the isolated compounds was determined using screening kits. Diclofenac sodium and celecoxib were used as standard drugs. The results of the enzyme inhibition assays (μM) and selectivity index (SI) are presented in Table 5. The in vitro results showed that the isolated compound showed potent activity against the COX-2 isozyme with an IC_50_ value of 9.08 ± 0.11 μM. However, it exhibited a poor inhibition potential against COX-1. The SI value for the compound was 4.85.

### 2.8. Molecular Docking

#### 2.8.1. Docking Studies on COX-1/COX-2

A molecular docking study of salicyalazine was conducted with the help of Molecular Operating Environment (MOE 2016) software. Three-dimensional 3D COX-1 and COX-2 isoforms were obtained from the Protein Data Bank (PDB). The accession code for the COX-2 isozyme co-crystallized with SC-558 was 1CX2. COX-1 is 3N8Y (diclofenac as the native ligand). Three-dimensional interaction plots of the salicyalazine in the binding sites of 1CX2 and 3N8Y are shown in Figure 4a,b. The isolated compound exhibited two π-sulfur interactions with Met113 and Met522. Tyr385 forms π-π interactions with the phenyl ring of the compound (Figure 4a). The compound showed significant hydrogen bond interactions with the key amino acid residues of the additional selectivity pocket residues (His90 and Gln192) of COX-2. Val523, which is a very important residue of the COX-2 additional pocket, forms π-σ interactions with the phenyl ring. The computed binding energy values for Salicyalazine in the binding pocket of COX-1 and COX-2 are −4.3845 kcal/mol and −6.0189 kcal/mol, respectively. For reference, we also docked the most potent COX-2 selective drug (Reference drug) into the binding sites of both isoforms of COX. The 2D dimensional interaction plots are shown in Appendix A. The computed binding energy values for celecoxib in the binding pocket of COX-1 and COX-2 are −5.3264 kcal/mol and −10.4291 kcal/mol, respectively.

#### 2.8.2. Docking Studies on μ-Opioid and GABA_A_ Receptors by Using MOE Software

Three-dimensional crystal structures of μ-opioid and GABA_A_ receptors were acquired from PDB with accession codes 5C1M and 4COF, correspondingly. The 3D ribbon diagram of salicyalazine into the binding site of μ-opioid (5C1M) is shown in Figure 5a. The 3D interaction plot of the purified salicyalazine showed a hydrogen bond interaction with Trp318. His54 forms π-π interactions with the phenyl ring. Hydrophobic interaction of the π-alkyl type was also observed (Figure 5b). 

Finally, the isolated compound was docked into the binding site of the GABA_A_ receptor (4COF). The surface diagram of the compound at the binding site of the enzyme is shown in Figure 6a. The compound displayed two hydrogen bond interactions with Asn41 as well as Gln64. Met115 formed a π-alkyl interaction (Figure 6b). The binding energy data of both COX isoforms are shown in Table 5. The 2D binding interaction poses of native co-crystallized ligands are μ-opioid, and GABA_A_ receptors are shown in Appendix A.

#### 2.8.3. Docking Studies on μ-Opioid and GABA_A_ Receptors by Using AutoDock Software

AutoDock 4.0 was also used for docking simulations. The purpose was conducting calculations of the inhibition constant (Ki) against the selected molecular targets. The computed binding free energies and inhibition constants (Ki) are presented in Table 6.

## 3. Discussion 

*Micromeria biflora* is an excellent medicinal herb that is widely used for treating toothaches, nose bleeding, wounds, and sinusitis [9]. The crude extract of *M. biflora* has excellent analgesic, sedative, muscle relaxant, and anti-inflammatory properties [10,11,12]. The promising antioxidant, antiglycation, antimicrobial, cytotoxic, and phytotoxic activities of *M. biflora* have also been documented [13]. The crude extract of *M. biflora* comprises active secondary metabolites responsible for the biological potential of the extract [9,13]. Therefore, the purification and structural identification of the natural products and subsequent screening of biological activity are necessary to identify the key compounds responsible for such biological properties. In this study, salicylalazine from *M. biflora* was assessed for its analgesic, muscle relaxation, sedative, and anti-inflammatory activities.

Prostaglandin (PG) is the main product of arachidonic acid, and the conversion is catalyzed by cyclooxygenase (COX). Most COX inhibitors are anti-inflammatory, and analgesic and have a strong side effect on the stomach [14,15]. Consequently, the development of new, effective, and non-toxic COX inhibitors is a major test for scientists. In the thermally induced screening test, salicylalazine exhibited a significant analgesic effect compared to the standard drug.

The muscle relaxation potential of salicylalazine was assessed in inclined plan and traction models, as per the standard procedure [16]. The results of the design investigation showed a strong potential of salicylalazine for the muscle relaxant effect. Salicylalazine showed excellent effects in both experimental models. In the inclined plan and traction models, the relaxation of muscle was studied after several intervals of time such as 30, 60, and 90 min of salicylalazine administration. An excellent effect was observed after 60 min of salicylalazine treatment at high doses.

Salicylalazine was also screened for sedative activity. Several natural products have been documented to have antihistaminic properties [16]. The antihistaminic effect might be related to H_1_ receptor blocking, leading to a sedative effect. In this study, salicylalazine proved to be an effective sedative.

Secondary metabolites with promising sedatives, analgesic, muscle relaxants, and anti-inflammatory activities have been documented. Based on this rationale, salicylalazine was screened for analgesia, muscle relaxation, sedative, and anti-inflammatory activities. Salicylalazine attenuated inflammation in carrageenan- and histamine-induced paw edema models. Carrageenan-induced screening involved two steps: the initial phase edema is endorsed to the local discharge of histamine, serotonin, and bradykinins, with the latter phase associated with the overproduction of PG [17,18]. The results showed that salicylalazine inhibited the biphasic edema induced by carrageenan, which is a sign that the purified constitutes a histamine blocker as well as a PG blocker. Thus, we recommend a more comprehensive and careful screening of salicylalazine as a lead compound to prevent various diseases. The sedative, analgesic, muscle relaxant, and anti-inflammatory effects of salicylalazine provide a strong scientific validation to the traditional use of *Micromeria biflora* for curing pain, muscle cramps, and inflammation. Salicylalazine isolated from *M. biflora* also exhibited excellent anticancer potential against NCI-H226, HepG2, A-4.98, and MDR2780AD cell lines. Based on these results, salicylalazine needs to be characterized by detailed clinical studies for the treatment of cancer.

Salicylalazine isolated from *M. biflora* was also screened for in vitro cytotoxicity using mouse hepatocytes and LCMK-2 monkey kidney cells. Salicylalazine did not exhibit any considerable cytotoxicity in the MTT assay, indicating the preliminary safety of salicylalazine. We performed docking studies on four targets. First, we studied the inflammatory pathway involving cyclooxygenase (COX) inhibition. Docking studies were conducted on both isoforms of COX (COX-1 and COX-2). In vitro enzyme inhibition results shown that isolated compound has potent activity against COX-2 isozyme with the IC_50_ value of 9.08 ± 0.11 μM and SI value of 4.85. The binding mode of the standard reference drug celecoxib revealed that it interacts with the amino acid residues present in the selectivity pocket of COX-2. The amino acid residues involved in the hydrogen bond interaction with celecoxib are His90, Gln152, Leu352, and Phe518. These strong hydrogen bond interactions are considered responsible for the high inhibition potential and selectivity of celecoxib. In our current study, salicylazine showed more selectivity towards COX-2. In the binding pocket of COX-2, it was oriented towards the selectivity pocket of COX-2 and established hydrogen bond interactions with His90 and Gln192. The strength of the ligand-enzyme complexes appeared as binding energy values. The GABAnergic system interacts with opioid receptors and muscarinic acetylcholine receptors and participate in nociception. Hence, we docked the compound on μ-opioid and GABA_A_ receptors. The compound under study showed a similar type of orientation in the binding sites of μ-opioid and GABA_A_ receptors, as shown by their respective native co-crystalized ligands. Next, we calculated the binding energy values using two software packages: MOE and AutoDock (v. 4.0). Moreover, we were interested in calculating the inhibition constant (K_i_) to explore the binding affinity. Autodock can calculate the Ki value using the binding energy. The isolated compound has a similar binding affinity toward GABA_A_ (Ki = 5.43 μM) and μ-opioid (Ki = 5.55 μM) receptors. The combination of COX/opioid/GABA_A_ receptors may be considered an effective pillar in multifaceted pain management by providing optimal therapeutic effects with few side effects.

## 4. Materials and Methods

### 4.1. Plant Collection

*Micromeria biflora* plant materials were collected from Dir, KPK, Pakistan. The specimen of the fresh obtained plant was recognized by Professor, Dr. Barkath Ullah (Department of Botany, Peshawar University KP, Pakistan). The marked specimen no UOP/Bot8826 was stored at the departmental herbarium.

### 4.2. Extraction and Isolation

The obtained plant materials were dried under shade for 17 days and then powdered using a local grinder machine. Among dried plant materials, 8.00 kg was assessed to cold extraction with polar solvent. The ground plant materials (8.00 kg) were soaked in a polar solvent (methanol) for 20 days and then concentrated with the help of a rotary evaporator. The obtained concentrate (167 gm) was subjected to Soxhlet extraction with hexane for fatty acid removal. Next, the defatted extract was assessed for chromatographic analysis using n-hexane and ethyl acetate as the solvent system, which afforded several sub-fractions. Sub-fraction MF-14 was assessed to a repeated chromatographic analysis by eluting the column with distilled n-hexane and ethyl acetate (100:0 to 13:87), yielding isolated salicylalazine (1.60 g). The structure of salicylalazine was identified by various spectroscopic techniques and compared with recently reported data [9,19].

### 4.3. Animals

Healthy mice (Balb/c) with weight ranging from 20 to 22 g and ages 9 to 10 weeks were used in this screening model. The grown animals were provided with standard laboratory conditions such as standard food and water ad libitum. The in vivo biological screening test was performed at the Department of Veterinary Medicine, College of Agriculture and Veterinary Medicine, Qassim University Buraydah, Saudi Arabia, as well as Swabi University, KP, Pakistan, according to published procedures.

### 4.4. Analgesic Activity

The analgesic effect of salicylalazine isolated from *M. biflora* was obtained with help of a hot plate according to the published method [20]. The animals were distributed into different groups according to the previously reported method [20]. A preliminary screening test was performed on the animals using a hot plate (Harvard apparatus) sustained at 55 ± 2 °C. The animals which exhibited more than 15 s, latency time were omitted from this screening. Group I (positive control) was administered with the standard drug, while Group II (negative control) was administered with normal saline. The other groups of animals were administered with 2.5, 5, 10, 15, and 20 mg/kg (PO). After 30 min of treatment, the screen animals were placed on the hot plate and kept at 55.12 °C; the latency time was defined as the time for which the animals were positioned on the hot plate at 55.12 °C without flicking and licking the hind limb, which was observed in seconds. To avoid tissue damage, a cut-off time of 30 s was used for every animal. The latency time was recorded for all groups (*n* = 6) at consistent intervals (30, 60, 90, and 120 min).

### 4.5. Muscle Relaxation

#### 4.5.1. Inclined Plane Model

An inclined plane screening procedure was adapted to identify the muscle relaxation effects of salicylalazine isolated from *M. biflora* according to a reported procedure [21,22]. The inclined plane used in screening contained two plywood boards, and these boards were connected, where one board was at 60 0 from the base while the other one was aligned from the base. Various groups of animals were distributed into several groups; each group comprised six animals (*n* = 6). Animals in each group were administered water (10 mL/kg), standard drug (diazepam, 1 mg/kg), and salicylalazine at different doses of 2.5, 5, 10, 15, and 20 mg/kg, i.p. After various time intervals, such as 30, 60, and 90 min post-administration, animals were allowed on the higher part of the inclined plane for 30 s to drop or hang.

#### 4.5.2. Traction Model

A traction screening model was adapted to identify the relaxation of muscle for salicylalazine isolated from *M. biflora* according to the standard method [21,22]. The traction model was considered by using metallic wires covered with rubber, and then the wires were linked with each other using a stand around 60 cm upstairs of the lab bench. The animals were distributed into different groups, with each group comprising six animals (*n* = 6). The various groups of animals were administered water, diazepam, and salicylalazine at different doses. After administration, all groups of animals were subjected to traction screening after 30, 60, and 90 min. After that, the animals were suspended by their back legs to hang from the connected wire, which demonstrated muscle relaxation potential. Traction testing was repeatedly performed to calculate the muscle relaxant ability of salicylalazine as per the reported method [22].

### 4.6. Sedative Activity

The sedative potential of salicylalazine isolated from *M. biflora* was assessed as per the reported procedure [22]. The animals were divided into different groups, as designated above. The apparatuses used in this screening contained an area of white wood with a 150 cm diameter bordered by stainless steel walls and was separated into 19 squares by black lines. Then, the open field apparatus was positioned inside the light and sound-reduced room. The different groups of animals were administered distilled water, diazepam, and salicylalazine at different doses. Thirty minutes after administration, all groups of animals were permitted to move from the center of the design box. The extreme number of lines crossed by animals was measured without sedation, while the animals that were considered sedated displayed delayed movement. Statistical calculations were performed by counting the number of lines crossed by every animal.

### 4.7. Anti-Inflammatory Activity

#### 4.7.1. Histamine Method

The anti-inflammatory potential of salicylalazine isolated from *M. biflora* was determined as per our recently published method [23]. The induction of inflammation was completed by administering 0.1 mL histamine (0.5%) to the animal’s sub plantar region. The volume of the paw was recorded 1–5 h after the administration of the inflammatory drug. Then, every group of animals was divided and pretreated orally with salicylalazine (15 and 20 mg/kg) for 1 h before making paw edema. The effect of the drug was related to that of loratadine (standard).

#### 4.7.2. Carrageenan Method

This model was used to identify the anti-inflammatory potency of salicylalazine purified from *M. biflora* according to the standard method [23]. The animals were randomly distributed into different groups depending on the treatment. Group I was treated with distilled water (10 mL/kg), which acted as a negative control. Group II was administered a standard drug (diclofenac sodium; 5 mg/kg), which acted as a positive control, while the other group was administered salicylalazine at 15 and 20 mg/kg. After 30 min of the intraperitoneal subjection of the mentioned administrations, 1% carrageen (0.05 mL) was intravenously administrated in the right paw of each animal. The inflammation in every animal was recoded after 1–5 h of carrageenan treatment using a plethysmometer (LE 7500 plan lab S.L). The anti-inflammatory action of salicylalazine was measured using the following formula:$$\mathrm{Activity}\;\left(\%\right)=\left(A-B\;/A\right)\times 100$$
where A is the paw edema of the control group, and B is the paw edema of the screening group.

### 4.8. Anticancer Assay Using MTT Method

The MTT assay was used to assess the cytotoxicity of salicylalazine isolated from *M. biflora* as per the previously documented method [23]. For the MTT assay, Gibco BRL was combined with 100 µg/mL of penicillin sodium salts, 10% FBS (Gibco), and sodium carbonate (2 mg/mL). The prepared medium was used to keep three human cancer cell lines: human hepatoma (HepG2), human A498, and renal NCI-h226 non-small cell lung). Mouse hepatocytes, including HepG2 and HepG-R, were planted in 96-well plates. The cell lines were preserved with salicylalazine (1.5–100 μM), hatched for 24 h, and then they were tracked using the MTT procedure^14^. This method was used for all tested cell lines. The IC_50_ value of salicylalazine in different cell lines was calculated from the concentration-effect curves. The standard drug used in this experiment was paclitaxel.

### 4.9. Cytotoxicity Assay

Salicylalazine isolated from *M. biflora* was also screened for in vitro cytotoxicity using mouse hepatocytes and LCMK-2 monkey kidney cells [23]. According to the MTT assay, salicylalazine was incubated for 24 h and the viability of the cell was identified. In this screening test, the cells were kept in RPMI 1640 medium as per the reported method. The medium used in this experiment consisted of streptomycin sulfate, NaHCO_3_ solution, and penicillin sodium salt, and FBS in proper proportion. The initial seeding of 8.6 × 10^3^ mice hepatocytes and 7.1 × 10^3^ LCMK-2 cells was indicated in 96 well plates. The cells were preserved with salicylalazine at different concentrations, and vehicle (0.2% DMSO), and hatching was performed for 48 h by MTT following our previously reported method.

### 4.10. Toxicological Screening

Salicylalazine isolated from *M. biflora* was also assessed for acute toxicity screening, which was divided into a salicylalazine-treated group as well as a saline-treated group accounting for the well-established method [24]. Salicylalazine was injected at different doses of 20, 50, 80, 100, and 200 mg/kg (P. O). After 30 min of administration of salicylalazine, every animal was observed cautiously for any gross effects for 8 h. The mortality rate was recorded for 24 h.

### 4.11. Molecular Docking

Docking simulations were performed to study the mechanism of salicylalazine isolated from *M. biflora* using MOE (2016.0802) and Autodock (*v* 4.0) software packages [25,26]. Docking simulations were performed on four molecular targets: cyclooxygenases (COX-1 PDB ID = 3N8Y and COX-2 PDB ID = 1CX2), GABA_A_ receptor (PDB ID = 4COF), and μ-opioid receptor (PDB ID = 5C1M).

Our previously reported methods were used to prepare the structures of all the co-crystalized ligands, isolated compound, and downloaded enzymes [10,27,28]. The rock method was used to validate the docking protocols. For this purpose, all the prepared structures were docked into the binding sites of their respective enzymes and their mode of binding in comparison with experimental ligands was analyzed via root-mean-square deviation (RMSD) computation. Only validated protocols with RMSD values < 2.0 Å were used for further studies. After docking studies, the 3D interaction plots were analyzed using Discovery Studio Visualizer and Chimera software [12].

### 4.12. Statistical Analysis

The results of in vivo biological activities are displayed as the mean ± standard mean (SEM) to fix the level of significant differences (*p* < 0.05 or *p* < 0.01) among all groups of animals.

## 5. Conclusions

*M. biflora* is traditionally used for treating toothache, nose bleeding, wounds, and sinusitis. Here, we show that compound 1 extracted from *M. biflora* has promising analgesia, muscle relaxation, sedative, anti-inflammatory, anticancer, and COX-1/2-inhibiting properties. Hence, our design study provides a scientific rationale for the folkloric usage of *M. biflora* for the cure of several disorders. Additionally, salicylalazine is an excellent candidate for further detailed studies to determine its clinical application. Docking studies were performed on four molecular targets of salicylalazine. The compound interacted with the selectivity-connected residues in the binding pocket of COX-2. In addition, the isolated compound has a similar binding affinity toward GABA_A_ (Ki = 5.43 μM) and μ-opioid (Ki = 5.55 μM) receptors. Therefore, salicylalazine isolated from *Micromeria biflora* may inhibit the COX-2/opioid/GABA_A_ pathways and may be considered an effective therapeutic agent for multifaceted pain management, with a few side effects.

## Figures and Tables

**Figure 1 molecules-27-03377-f001:**
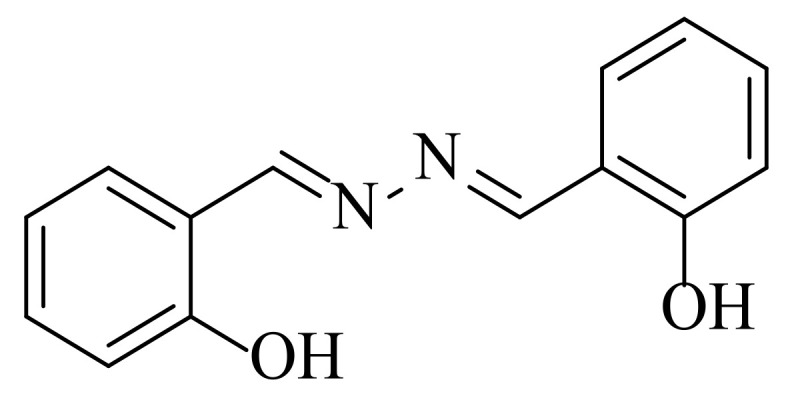
Molecular structure of salicylalazine purified from *M. biflora*.

**Figure 2 molecules-27-03377-f002:**
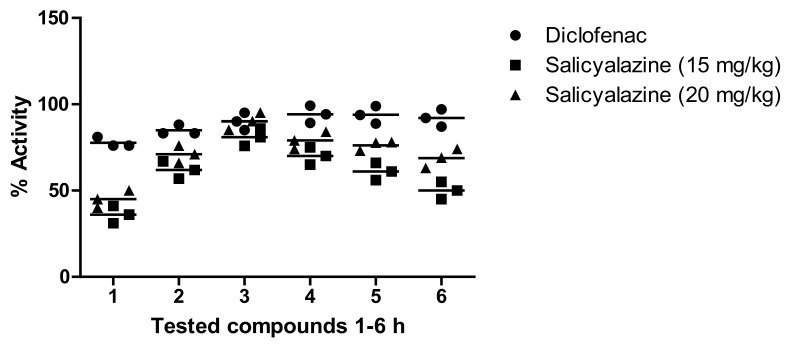
Anti-inflammatory potential of salicyalalzine (15, 20 mg/kg) of the *M. biflora* on carrageenan paw in mice. The obtained results are shown as ± S.E.M for six groups of animals. In the histamine-induced paw edema model, salicylalzine (15 mg/kg) showed 78.45% and 74.33% effects in the initial phase (at 1 and 2 h time points) and was continued up to the second phase (58.09%) at 3 h. An excellent effect (90.55% and 93.99%) was observed at 20 mg/kg at the 1 h and 2 h time points, respectively, while it reached 86.11% after 3 h (Figure 3).

**Figure 3 molecules-27-03377-f003:**
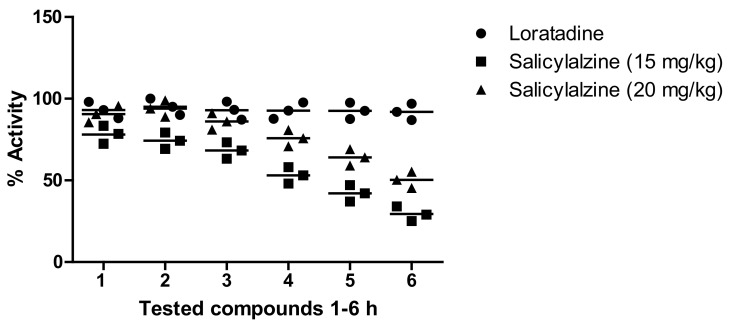
Anti-inflammatory potential of salicyalalzine (15, 20 mg/kg) of the *M. biflora* on histamine-induced paw in mice. The obtained results are shown as mean ± S.E.M for six groups of animals.

**Figure 4 molecules-27-03377-f004:**
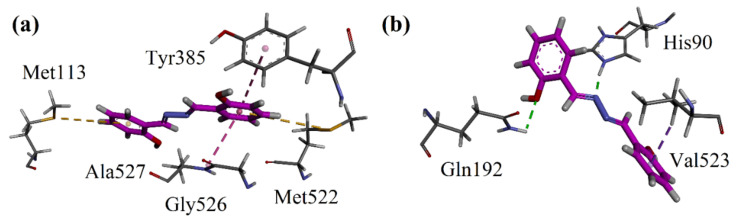
Close-up 3D interaction plot of salicyalazine in the binding site of (**a**) COX-1 and (**b**) COX-2.

**Figure 5 molecules-27-03377-f005:**
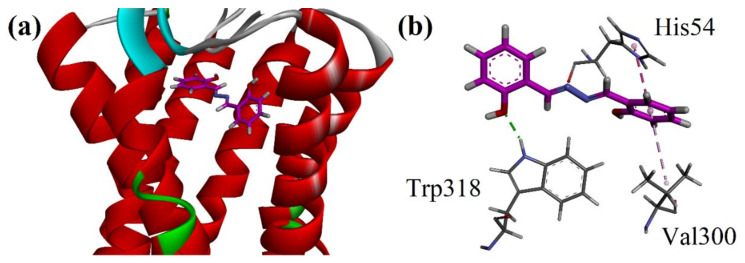
(**a**) The 3D ribbon diagram of isolated salicylalazine into the binding site of μ-opioid (5C1M); (**b**) 3D interaction plot in the binding site of 5C1M showing interactions with key amino acid residues.

**Figure 6 molecules-27-03377-f006:**
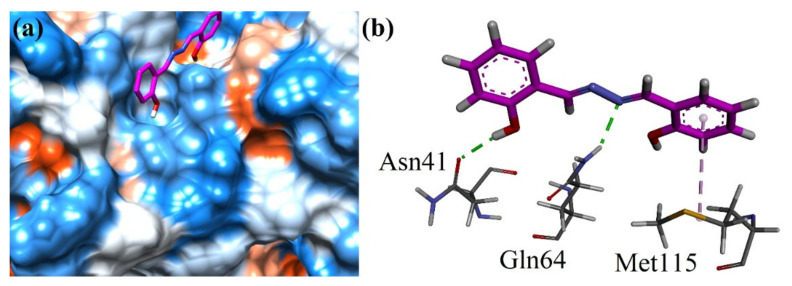
(**a**) The 3D surface diagram of isolated salicylalazine into the binding site of GABA (4COF). (**b**) Close-up 3D interaction plot of purified salicylalazine in the binding site of 4COF.

**Table 1 molecules-27-03377-t001:** Analgesic effect of salicylalazine isolated from *M. biflora*.

Group	Dose mg/kg	Time in Minutes			
		30	60	90	120
Normal Saline	10 mL	9.22 ± 0.09	9.21 ± 0.09	9.22 ± 0.08	9.18 ± 0.10
Tramadol	10	24.30 ± 0.07 ***	25.90 ± 0.08 ***	24.80 ± 0.14 ***	24.68 ± 0.40 ***
Salicylalazine	2.5	13.33 ± 0.55 *	14.40 ± 0.83 *	14.33 ± 0.64 *	14.01 ± 0.43 *
	5	15.12 ± 0.80 **	18.09 ± 0.69 **	18.03 ± 0.90 **	17.97 ± 1.00 **
	10	18.65 ± 0.95 **	20.89 ± 0.90 ***	20.77 ± 0.87 ***	19.97 ± 0.80 **
	15	21.43 ± 0.50 ***	22.89 ± 0.80 ***	22.00 ± 0.98 ***	21.65 ± 0.98 ***
	20	23.00 ± 0.44 ***	23.90 ± 0.97 ***	22.80 ± 1.05 ***	22.79 ± 1.09 ***

* *p* < 0.05; ** *p* < 0.01; *** *p* < 0.001.

**Table 2 molecules-27-03377-t002:** Muscle relaxation potential of salicylalazine purified from *M. biflora*.

Group	Dose (mg/kg)	Inclined Plane Test (%)			Traction Test (%)		
		30 min	60 min	90 min	30 min	60 min	90 min
Distilled water	10	0.00 ± 0	0.00 ± 0	0.00 ± 0	0.00 ± 0	0.00 ± 0	0.00 ± 0
Diazepam	1	100 ± 0.00	100 ± 0.00	100 ± 0.00	100 ± 0.00	100 ± 0.00	100 ± 0.00
Salicylalazine	2.5	20.11 ± 1.45	26.23 ± 1.66	27.45 ± 1.23	21.23 ± 1.55	27.78 ± 2.10	28.65 ± 1.02
	5	27.34 ± 1.66	33.09 ± 1.60	34.09 ± 1.30	28.12 ± 1.60	34.56 ± 2.07	35.34 ± 2.00
	10	33.98 ± 1.40	38.08 ± 1.55	39.43 ± 1.28	34.99 ± 1.70	39.45 ± 2.12	40.66 ± 1.88
	15	44.53 ± 1.43	50.94 ± 1.50	51.52 ± 1.20	45.78 ± 1.88	52.98 ± 2.15	52.32 ± 2.04
	20	52.09 ± 1.30	59.34 ± 1.45	60.88 ± 1.19	53.72 ± 1.89	60.56 ± 2.13	61.32 ± 2.06

**Table 3 molecules-27-03377-t003:** The sedative potential of salicylalazine purified from *M. biflora* in the open field test.

Sample	Dose mg/kg	No of Lines Crossed
Control	5 mL	129.29 ± 3.88
Diazepam	0.5	9.01 ± 0.51 ***
Salicylalazine	2.5	78.54 ± 0.98
	5	69.98 ± 1.22
	10	60.90 ± 1.00 **
	15	51.09 ± 0.97 ***
	20	40.98 ± 0.80 ***

** *p* < 0.01; *** *p* < 0.001.

**Table 4 molecules-27-03377-t004:** In vitro anticancer effect of salicylalazine isolated from *M. biflora*.

Tested Compound	IC50 μM			
	HepG2	A498	NCI-H226	MDR2780AD
Salicylalazine	20.76 ± 0.38	115.54 ± 0.42	64.98 ± 0.11	0.85 ± 0.09
Paclitaxel	7.51 ± 0.30	95.23 ± 0.23	61.29 ± 0.31	0.18 ± 0.008

**Table 5 molecules-27-03377-t005:** In vitro COX1/2 inhibition of salicylalazine isolated from *M. biflora*.

Compound	IC_50_ (μM)		SI
	COX-1	COX-2	
Salicyalazine	44.11 ± 1.18	9.08 ± 0.11	4.85
Diclofenac sodium	0.18 ± 0.01	0.09 ± 1.02	2
Celecoxib	6.51 ± 1.21	0.024 ± 0.001	271

SI = IC_50_ (COX-1)/IC_50_ (COX-2).

**Table 6 molecules-27-03377-t006:** Binding energy and inhibition constant of the salicylalazine calculated by Autodock into the binding site of μ-opioid and GABA_A_ receptors.

GABA_A_ Receptor (PDB ID = 4COF)			μ-Opioid Receptor (PDB ID = 5C1M)		
MOE	Autodock		MOE	Autodock	
BE (kcal/mol)	BE (kcal/mol)	Ki (μM)	BE (kcal/mol)	BE (kcal/mol)	Ki (μM)
−6.9951	−7.17	5.55	−6.8406	−7.18	5.43

## Data Availability

The data associated with this study are available in the main text of this paper.

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
