# Peer review of "In Vivo and In Vitro Biological Evaluation and Molecular Docking Studies of Compounds Isolated from Micromeria biflora (Buch. Ham. ex D.Don) Benth"

_molecules, 2022, doi:10.3390/molecules27113377_

Round 1

Reviewer 1 Report

The authors presented ‘In Vivo and In vitro Biological Evaluation and Molecular 2 Docking Studies of Compound Isolated from Micromeria 3 biflora (Buch. Ham. ex D.Don) Benth’. Series of vivo and vitro bioactivities of Salicyalazine compound were investigated in this work. The work strategy of this manuscript is classical, however the authors evaluated different bioactivities (such as of analgesic, muscle relaxation, sedative, anti-Inflammatory, cytotoxicity effects and inhibition of cyclooxygenases) to show the correlation between traditional medicine of the Micromeria biflora plant and the deep scientific research on this plant. In particular, Salicyalazine is one of the most bioactive compounds of this plant. A new contribution in this work is that the molecular docking was indicated, showing the interaction of Salicyalazine with the protein system.

I have some comments that could help other authors tremendously and also make the current manuscript an encyclopedia-type publication, which would benefit both the authors and the readers.

  • Even the Salicyalazine compound is known, I suggest the authors briefly give the structure determination of this compound.
  • The bioactivity of compound can influence their bioactivity, please indicate the purity of salicyalazine being isolated and the method for the purity determination.
  • For all tests of in vivo and vitro, the authors should clearly indicate the reference compounds used in this study and compare the activity level of Salicyalazine with those of the reference compounds.
  • Several bioactivities of Salicyalazine were mentioned in this manuscript. However, the authors should discuss the relationship of structure-activities.
  • Does another compound with similar structure to Salicyalazine have bioactivity, as mentioned in the previous publications.

Author Response

The authors presented ‘In Vivo and In vitro Biological Evaluation and Molecular 2 Docking Studies of Compound Isolated from Micromeria 3 biflora (Buch. Ham. ex D.Don) Benth’. Series of vivo and vitro bioactivities of Salicyalazine compound were investigated in this work. The work strategy of this manuscript is classical, however the authors evaluated different bioactivities (such as of analgesic, muscle relaxation, sedative, anti-Inflammatory, cytotoxicity effects and inhibition of cyclooxygenases) to show the correlation between traditional medicine of the Micromeria biflora plant and the deep scientific research on this plant. In particular, Salicyalazine is one of the most bioactive compounds of this plant. A new contribution in this work is that the molecular docking was indicated, showing the interaction of Salicyalazine with the protein system.

I have some comments that could help other authors tremendously and also make the current manuscript an encyclopedia-type publication, which would benefit both the authors and the readers.

  • Even the Salicyalazine compound is known, I suggest the authors briefly give the structure determination of this compound.

Reply: The chemical structure determined by using NMR and mass spectrometer is being included briefly now.  

  • The bioactivity of compound can influence their bioactivity, please indicate the purity of salicyalazine being isolated and the method for the purity determination.

Reply: The purity of salicyalazine is included now.

  • For all tests of in vivo and vitro, the authors should clearly indicate the reference compounds used in this study and compare the activity level of Salicyalazine with those of the reference compounds.

Reply: Thank you for this attention. Actually all the activities were performed as negative control, positive control (reference drug) and samples to be tested. Now all the groups were compared with negative control for finding the level of statistical significance. 

  • Several bioactivities of Salicyalazine were mentioned in this manuscript. However, the authors should discuss the relationship of structure-activities.

Reply: there is a significant correlation among the conducted activities. Specially the analgesic, muscle relaxant and anti-inflammatory. While the sedative effect is also related with muscle relaxant like diazepam/ oxazepam are the famous sedative and muscle relaxant.   

  • Does another compound with similar structure to Salicyalazine have bioactivity, as mentioned in the previous publications.

Reply: There is no structure activity relationship on the said compound.

Reviewer 2 Report

This paper is probably publishable, but should be reviewed again in revised form before it is accepted.

  • I did not find information in methodology section how/if the ligand molecules were optimized. It should be standard procedure before docking.
  • My complaint is also I could not find information about validation of docking procedure. Authors mentioned previous studies but did not provide any references to them.
  • It is well known that scoring functions which are used in the docking algorithms only give approximate values of binding energies and inhibition constants. More valuable is an illustration of structural basis of inhibition and its comparison with the binding poses of native ligands/drugs (collected from protein data base). The Figure which compare the mode of binding of compound and control molecules should be presented and discussed in the main text.
  • In the methodology section, the names of proteins retrieved from the database were mixed up. It should be corrected.
  • Why μ-opioid and GABAa were chosen for docking.Choosing COX isoforms seems reasonable.
  • The manuscript requires extra editing for English language and phrasing in some paragraphs before being accepted for publication.

Author Response

This paper is probably publishable, but should be reviewed again in revised form before it is accepted.

  • I did not find information in methodology section how/if the ligand molecules were optimized. It should be standard procedure before docking.

Reply: Thank you very much for your comment. Actually, we have performed all the docking protocols (Energy minimization of the ligand, preparation of structures of the downloaded enzymes and active site identification) by using our previously reported methods (References 27-29). were carried out according to a previously reported procedure.

  • My complaint is also I could not find information about validation of docking procedure. Authors mentioned previous studies but did not provide any references to them.

Reply: Thank you very much for your comment. In the original manuscript, we have already described the validation of docking protocol. We have highlighted with green in the revised version.

  • It is well known that scoring functions which are used in the docking algorithms only give approximate values of binding energies and inhibition constants. More valuable is an illustration of structural basis of inhibition and its comparison with the binding poses of native ligands/drugs (collected from protein data base). The Figure which compared the mode of binding of compound and control molecules should be presented and discussed in the main text.

Reply: Thank you very much for your insightful comment. In the revised version, we have compared the binding orientation of native or reference drug with the compound under study. It is shown in Results as well as in Discussion section. While their binding orientations are presented in Supporting Information file. The compound showed almost same binding orientations and interacts with key amino acid residues.

  • In the methodology section, the names of proteins retrieved from the database were mixed up. It should be corrected.

Reply:  Thank you very much for comment. It as been corrected in revised version.

  • Why μ-opioid and GABAa were chosen for docking. Choosing COX isoforms seems reasonable.

Reply: GABA is responsible for muscle relxant and sedation while COX is related to analgesia.

  • The manuscript requires extra editing for English language and phrasing in some paragraphs before being accepted for publication.

Reply: In revised version, special attention has been given to improve the English language and phrasing.

Round 2

Reviewer 1 Report

The authors has revised the manuscript as suggestions.